# Maximizing Quality of Life in Children with Epilepsy

**DOI:** 10.3390/children10010065

**Published:** 2022-12-28

**Authors:** Yu-Tze Ng

**Affiliations:** Department of Child Health, University of Missouri School of Medicine, 400 N Keene St, Columbia, MO 65201, USA; ytng@health.missouri.edu; Tel.: +1-(573)-882-6882

**Keywords:** children, stigma of seizures, parents, QOL

## Abstract

Arguably significant progress and improvement in the medical and surgical treatments of seizures and epilepsy in children have occurred; however, there have been relatively fewer efforts in optimizing the care of lifestyle complications related to the disease state. Many patients have significant behavioral and mental health comorbidities, including ADHD (attention deficit hyperactivity disorder), which should be treated. After epilepsy surgery, only seizure freedom results in improved quality of life (QOL). Improved compliance leads to better seizure control and ensuring that caregivers have a rescue treatment helps empower patients. Education and improving seizure illness perception is beneficial. Cannabidiol may have benefits other than seizure control. The majority of children are mainly concerned about the stigma attached to having epilepsy. Driving affecting older children is discussed. Restrictions on these children should be minimized and enabling regular activities maximized.

## 1. Introduction

We have previously shown that despite the many comorbidities and potential risks of seizures, epilepsy, and its treatments, the majority of children are most concerned with the stigma attached to having epilepsy [1]. The far-reaching negative impact of having epilepsy often includes social, psychological and financial complications beyond the medical and physical aspects. Quality of life (QOL) is defined as an individual’s perception of physical health, psychological state and level of independence, social relationships and personal beliefs [2]. There have been efforts to stress improving quality of life in children with epilepsy but it is my opinion that often there are no clear directives on how that can be achieved. I attempt to recommend concrete guidelines to attaining this goal, discussing QOL studies in children.

## 2. Rescue Treatments

Caregivers expressed feelings of helplessness and inability to do anything when their child had a seizure as amongst the worst features of epilepsy [1]. It is fairly simple to ensure that patients’ families are empowered with a rescue treatment. Once prescribed, at follow-up consultations, it is helpful to check and remind families to carry the medication everywhere they go, as often the drug is not on them. Hence with effective rectal diazepam treatment, apart from the concerns of embarrassment and other difficulties with administration, the tube itself is of significant size and often not carried everywhere by caregivers. Clonazepam oral dissolving tablets and wafers have been shown to be as effective as rectal diazepam in relieving seizures and also have the added benefit, other than a more socially acceptable route of administration, of being a very small package that is easily carried [3]. Indeed, I have had fathers proudly show their carrying of the tablet in their wallets. Recently, intranasal midazolam has been FDA-approved for the treatment of acute repetitive seizures and in the near future there will likely be available a buccal film of diazepam. Regardless of the formulation and type of benzodiazepine, the major factor is ensuring it is always by the patient and available. An advantage to the vagus nerve stimulator (VNS) other than the fact that one does not have to remember to do anything (once the device has been programmed) is that it also offers the patients and their caregivers something to do on the event of a seizure. Placing and removing the magnet over the device at the onset of the seizure aborts some of the seizures some of the time. The “additional” benefits of the VNS are real, often making the patients feel happier (it is FDA-approved for refractory depression) and more alert, and we have previously described the likely related initial development of speech [4]. Subsequently the mother of a child with autism reported that at age two, she completely stopped talking. Nearly a decade later, following implantation of the VNS, she started speaking in sentences the moment it was turned on [personal communication]!

## 3. Medication Compliance

There has been some criticism of pharmaceutical companies simply developing extended-release formulations of common antiepileptic drugs, instead of investing in novel compounds and treatments for epilepsy. An example are two brand formulations of topiramate, i.e., Qudexy XR and Trokendi XR. However, missing medication doses remains the commonest cause of breakthrough seizures in patients with epilepsy and related subsequent complications. It has been demonstrated that once-daily and less frequent dosing results in better compliance [5]. Hence, every little bit of maximizing the ease of administration and compliance of medications is useful in improving the care of these patients. A potential least compliant group of patients is the oxymoron, “mature teenagers”. This group of patients, unlike toddlers, mainly do not voluntarily fight to refuse the medications, spitting it out, etc. They are simply too busy and distracted with their lives to bother remembering and their parents often have the mistaken idea that they will be responsible enough to take their medications because they appropriately do not want to have any seizures. There are several simple solutions for this:Ensure detailed parental supervision of them actually taking their medications every time, at least for a couple of weeks or months initially.Like us, teenagers are never away from their phones, so set the daily alarm time for them to take their usually nighttime-dosed medication.Obtain a pillbox and use it.

## 4. Mental Health

A cross-sectional equation model of baseline QUALITE cohort study, which included six Canadian child-epilepsy ambulatory programs, came to the following conclusion: mental health and social supports, not their seizures, are strongly related to epilepsy-specific quality of life in children aged 8 to 14 years of age [6]. This was a study of 480 children aged between 8 to 14 years of age with a receptive vocabulary of at least 70 in which the main outcome measure was QOL, using self-report measures with the Child Epilepsy QOL questionnaire [7]. From the perspective of these children, QOL was associated with mental health and peer and parental support, not seizure severity. This finding is similar to literature reviews that showed that repeated positive daily experiences (such as the family or at school) are more important to children’s perceived life satisfaction than even a major stressor, such as the diagnosis of epilepsy [6]. In contrast, caregiver reports of children’s QOL highlight the importance of cognition and seizure factors, once again showing differences in perception compared to the children themselves. A Korean study of 93 parents of a child with epilepsy using the Korean versions of the QOL in Childhood Epilepsy questionnaire (K-QOLCE), Parenting Stress Index (PSI) and Family Management Measure (FaMM) showed easy family-management styles were the most important factor in predicting QOL in childhood epilepsy patients. They recommended comprehensive intervention programs for parents and families of these children to promote positive perceptions of the child’s life and increase the parental management ability of the child’s condition and parental mutuality [8].

There are numerous papers that highlight illness perceptions determine psychological distress and QOL in patients with epilepsy in both children and adults. Illness perceptions can be defined as patients’ beliefs and expectations about their illness, whilst psychological distress is mostly defined as a state of emotional suffering characterized by symptoms of depression and anxiety [9]. A Greek study of 100 children aged 10–18 years with a mean age of 13.9 years found a better QOL was reported by those who believed that treatment did not control their illness and that their epilepsy would not affect them emotionally. Expectedly, the children who expected their disease will last a long time, who believe they have less personal control over their illness and who expect their epilepsy to have a high emotional impact reported higher levels of distress. Hence, the authors concluded that improving illness perceptions should be targeted as interventions in children with epilepsy. In view of intervention, however, it is important that individual patients’ illness perceptions be screened at intake, as they (illness perceptions) may differ significantly from patient to patient [9]. It was not clearly stated how best illness perceptions could be improved, although, in general, education about the disease state being delivered to the patient and caregivers appears to help.

Mental health and social supports are strongly related to the QOL of children with epilepsy. A QOL survey study of 480 children (aged 8–14 years, and with receptive vocabulary scores of at least 70) with epilepsy from six Canadian child-epilepsy outpatient clinics was performed using the Child Epilepsy QOL questionnaire [10]. Epilepsy-specific QOL from the child’s perspective is strongly related to their mental health and social support but not their seizures. Both peer and parental support exhibited direct associations with QOL. Estimated verbal intelligence exerts its strongest association with QOL through mental health, whereas seizure status has a weak relationship to QOL only through mental health, although there was selection bias in excluding 26 children with low vocabulary scores. The authors concluded controlling seizures is insufficient care for influencing the children’s QOL [6].

## 5. ADHD

A Turkish study of 35 children with epilepsy and attention deficit hyperactivity disorder (ADHD) compared them to 53 children with epilepsy and 52 children with primary ADHD [10]. The QOL scores were lowest in the epilepsy-ADHD group, even though their ADHD drug treatment rates were very similar—80.8% primary ADHD and 78.1% for epilepsy-ADHD. These authors concluded that epilepsy, with its social stigma, chronic course, learning problems and fear of seizures, and the chronic inattentiveness, hyperactivity and impulsivity symptoms of ADHD, with pervasive negative impact on the social, family and academic life, both led to poorer QOL. The co-occurrence of these two conditions could result in an increased risk of psychosocial and cognitive problems and a broader impairment of QOL [10]. Although both groups of children with ADHD had perhaps surprisingly similar rates of treatment with simulant and atomoxetine treatments, the epilepsy-ADHD group of children still had lower QOL. I think it is still very important to treat the ADHD whenever warranted. I am aware of the concerns of the general medical and even neurology communities about the theoretical risk of ADHD medications increasing the frequency of seizures and epilepsy, although there is evidence showing there is no increase in seizure frequency with stimulant use [11,12]. I do not believe there is any increase in seizure frequency with stimulant medications, but *even if there were* a slight, e.g., 10%, exacerbation of seizures, as long as the medications are clearly helping the child’s ADHD symptoms, we should be able to manage the epilepsy better, optimizing the antiepileptic drug regimen as appropriate.

There are likely neurobiological bases of ADHD-epilepsy comorbidity, focused on excitation/inhibition balance of these two conditions. Research has been performed on the identification of the molecular and cellular determinants of both these disorders as well as other neuropsychiatric disorders including even autism spectrum disorder and migraine. This (translational) research has shown us there are multiple mechanisms and molecules involved including biochemical agents, neuroinflammation and even mitochondrial dysfunction [13,14,15].

## 6. Cannibanoids

An open-label study of 48 young patients (median age 11.7 years) treated with purified cannabidiol (Epidiolex; GW Pharmaceuticals) was performed using the Quality of Life in Childhood Epilepsy (QOLCE) survey at the pretreatment visit and after 12 weeks of treatment [16]. The baseline mean overall QOLCE score was 37.8 ± 7.8, which increased to 45.7 ± 8.5 (*p* < 0.001) following treatment. These results were independent of changes in seizure frequency with no significant difference between the responders (>50% seizure reduction) and non-responders. Significant improvements in multiple QOLCE domains including energy/fatigue, memory, other cognitive functions, control/helplessness and social interactions and behavior, as well as improvements in general QOL subscores, were reported. The authors acknowledge the primary limitations of the study were the lack of a blinding or comparator group [16]. Whether in reality or through the power of belief or placebo, I have noticed many reported QOL improvements from our patients treated with cannabidiol, both from a Texas dispensary (or other source) and with Epidiolex; examples include being able to hold their bottle and feed themselves and being much more alert and aware. Nearly always, even when there has been no improvement in their seizure control, the caregivers want to continue cannabidiol due to its real or perceived benefits.

## 7. Driving

Driving remains a contentious issue [17,18]. The duration a licensed river with epilepsy has to be seizure-free varies usually between 3 to 12 months in the various states in the U.S., and is often longer in other countries. This is a necessary restriction, although the argument has been made that if we are banning patients with epilepsy from driving because they are considered dangerous, then we should ban all teenage male drivers. It is also known that in states that mandate physician reporting of their patients’ seizures, their patients start withholding information (lying) to their providers. The advent of widely available private transportation services, e.g., Uber and Lyft, and near-future independent self-driving cars have arguably made people who do not drive or own cars more independent. Similarly, many young people these days do not have a strong desire to obtain a driver’s license as early as possible. I counsel my patients that it is probably better to be a little delayed in obtaining their driver’s license (>3 months seizure free in Texas) than to obtain one then lose it with a breakthrough seizure, by which time they will have experienced the independence of driving but are then punished by having their driver’s license revoked.

## 8. Minimizing Restrictions

Optimizing medical and other treatments for seizure control and QOL are important tasks for medical providers. However, stressing the “normalization” of our patients with epilepsy and helping enable them remain of primary importance. Restrictions should be minimized; patients and families often ask about diet, sleep and teenage activities, e.g., sleeping overnight with friends. Apart from obvious precautions, such as swimming with lifeguard supervision and strict observation in water, on busy roads or at heights, everything else should be allowed, because by restricting their activities, diet, sleep regimens, etc., these children are in fact being punished for having epilepsy [19]. I allow my patients to play football and box. Football is a dangerous game, as is boxing; hence, if the authorities wanted to ban these sports, that would be understandable (although I do not recommend such drastic measures), but I do not discriminate against my patients with epilepsy and allow them the privilege of participating in risky sports. My approval letters include caveats such as: “Johnny is *cleared* to play football; however, football is a dangerous game and having seizures and epilepsy does not protect one from injuries”. Although I am not under any delusions that such an approval note will protect anyone from responsibility or litigation, it does make me feel better. I also encourage my older patients to stay with friends overnight and party like “normal teenagers”, and stop just short of telling them it is okay to experiment with drugs, because they are already being punished by having epilepsy and do not need any more punishment. Spreading the word about minimizing restrictions, including educating school nurses and staff, is of paramount importance. Their calling 911 and ordering emergency medical services as well as often demanding that a parent of the child come take them home from school after an often brief seizure is far more disruptive to their lives than the actual epilepsy!

## Data Availability

This study does not contain any original research data.

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
