# Peer review of "Maximizing Quality of Life in Children with Epilepsy"

_children, 2022, doi:10.3390/children10010065_

Round 1
Reviewer 1 Report
Manuscript ID: children-2095064-peer-review-v1
Yu-Tze in the present commentary article entitled ‘Treating More than just The Epilepsy; Maximizing Quality of Life’ provides his viewpoint on a timely topic, the improvement of Quality of Life in children with epilepsy, that often may show significant behavioral and mental health comorbidities, including ADHD.
The main strength of this manuscript is that it addresses an interesting and relevant question, providing concrete guidelines to ensure medical and other treatments (like playing sports or working on the stigma attached to having epilepsy) for seizure control and QOL improvement. In general, I think the idea of this paper is really interesting and the author’s fascinating observations on this timely topic may be of interest to the readers of Children.
Still, I have some comments to do, that in my opinion will help in improving the quality of the manuscript, its adequacy, and its readability prior to the publication in the present form.
First of all, I would suggest to modify the title that, in my opinion, is too long and appear to be unfocused and not so representative of the contents of the article. Also, as the author based its argumentation on the necessity of QOL improvement by discussing studies that have explored pharmaceutical and behavioral measures to reduce epilepsy and to assess psychological distress and QOL in both patients with epilepsy and in their parents, I believe that it could be interesting to discuss here neurobiological bases of ADHD-epilepsy comorbidity, focusing on excitation/inhibition balance that may provides more information on the identification of the molecular and cellular determinants of both these disorders (https://doi.org/10.3389/fnbeh.2022.946263; doi: 10.3390/biomedicines10010076; https://doi.org/10.3389/fnbeh.2022.998714; https://doi.org/10.3390/cells11162607). Finally, the author should consider revising the bibliography, as there are several incorrect citations. Indeed, according to the Journal’s guidelines, he should provide the abbreviated journal name in italics, the year of publication in bold, the volume number in italics for all the references. Also, please correct in-text citations: reference should be numbered, and placed in square brackets [ ] (for example [1]).
I hope that, after these careful revisions, the manuscript can meet the Journal’s high standards for publication. I am available for a new round of revision of this article.
I declare no conflict of interest regarding this manuscript.
Best regards,
Reviewer
Reviewer 2 Report
1. The abstract talks more about children and their caregivers, while the text contains information about the patient's driving, there is no coordination between the text, the title, and the abstract.
2. A large part of the text is based on the author's opinion and it is not based on the existing documents and articles.
3. The sentence start with “Subsequently the mother” should be cited appropriately.
4. There was no logical relationship between the drug treatment and quality of life in the introduction section.
5. This sentence start with “Apart from obvious precautions such as swimming with” should be cited appropriately.
Author Response
Please see the attachment (same as for Reviewer 1).

Round 2
Reviewer 2 Report
The most concerns have been addressed.